# Morphology and Molecular Phylogeny of *Fuscheriides baugilensis* sp. nov. (Protozoa, Ciliophora, Haptorida) from South Korea †

Seok Won Jang [1] , Atef Omar [2,3] , Seung Won Nam [1] and Jae-Ho Jung [4,*]

1 Protist Research Team, Nakdonggang National Institute of Biological Resources, Sangju 37242, Korea; jangsw1007@nnibr.re.kr (S.W.J.); seungwon10@nnibr.re.kr (S.W.N.)
2 Natural Science Research Institute, Gangneung-Wonju National University, Gangneung 25457, Korea; ganeb2002@yahoo.com
3 Department of Zoology, Al Azhar University, Assiut 71524, Egypt
4 Department of Biology, Gangneung-Wonju National University, Gangneung 25457, Korea
* Correspondence: jhjung@gwnu.ac.kr
† LSID [urn:lsid:zoobank.org:pub:A163BA95-3A25-4147-8BD0-1EF97E5C2476].

**Abstract:** The morphology and molecular phylogeny of a new haptorid ciliate, *Fuscheriides baugilensis* sp. nov., discovered in a temporary pond in South Korea, were investigated. The new species is characterized by its small body size (30–55 × 15–20 μm in vivo), oblong to rod-shaped extrusomes in the oral bulge and cytoplasm, 14–16 somatic kineties, two dorsal brush rows, and single subapical ciliary condensation. The phylogenetic analyses based on the 18S rRNA gene sequences show that the family Fuscheriidae is paraphyletic and the species belong to the genera *Fuscheriides* and *Pseudofuscheria* cluster together in the same subclade, while *Fuscheria* is in a different subclade, suggesting that the subapical ciliary condensation characterizing the two former genera has a higher taxonomic value than the shape of extrusomes for genera separation.

**Keywords:** Acropisthiina; Fuscheriidae; *Pseudofuscheria*; SSU rDNA; taxonomy





## 1. Introduction

The haptorid family Fuscheriidae Foissner et al., 2002 is comprised of predatory ciliates feeding on other protists, such as ciliates and flagellates and inhabiting freshwater and terrestrial habitats [1,2]. The fuscheriid ciliates belong to the suborder Acropisthiina Foissner and Foissner, 1988 [3], that is, their nematodesmal bundles originate from both the circumoral dikinetids and the oralized anterior somatic monokinetids and consist of three families, namely Acropisthiidae Foissner and Foissner, 1988, Fuscheriidae, and Pleuroplitidae Foissner, 1996. However, the family Fuscheriidae is similar to the family Pleuroplitidae in that they have meridional and not curved anteriorly somatic kineties and differ mainly in the absence vs. presence of a subapical, extracytostomal extrusome bundle on the ventral side. On the other hand, both families can be separated from the family Acropisthiidae [3] in that the latter has a spathidiid general organization with somatic kineties that are more or less curved anteriorly. The family Fuscheriidae comprises ten genera based on three main features: the number of dorsal brush rows; the shape of the extrusomes; and the presence/absence of the subapical ciliary condensation [1,2,4–7].

During the last decade, the SSU rRNA gene sequences of only two *Fuscheria* species, an unidentified *Fuscheria* sp., an unidentified *Fuscheriides* sp., and *Pseudofuscheria terricola* were added to the GenBank database [8–11]. The lack of molecular data from most of the fuscheriid genera makes it difficult to test the phylogenetic significance of the three main generic characters used to differentiate between these simply organized taxa. In the present study, we investigate the morphology of a new species discovered in a temporary puddle, South Korea. This species agrees very well with the diagnostic features of the genus

*Fuscheriides* provided by Gabilondo and Foissner [2], that is, with oblong to rod-shaped extrusomes, two dorsal brush rows, and a subapical ciliary condensation. Furthermore, the 18S rRNA gene sequence was analyzed to determine the phylogenetic position of the new species.

## 2. Materials and Methods

### 2.1. Sample Collection and Identification

*Fuscheriides baugilensis* sp. nov. was isolated from a water sample collected from a temporary puddle on a footpath (Baugil) behind the Gangneung-Wonju National University, Gangneung-si, South Korea in June 2020. The sample was transported to the laboratory, and a raw culture was established in a Petri dish at room temperature with the sterilized rice grains as a food source. Living cells were examined under a stereomicroscope (Olympus SZ61, Tokyo, Japan) and light microscope (Olympus BX53) with a differential interference contrast at magnifications of 50–1000×. The infraciliature was revealed by protargol and silver carbonate impregnation methods. The protargol powder was synthesized using the method of Kim and Jung [12]. The cells were fixed using concentrated Bouin's solution [13], and the protargol impregnation technique is based on 'procedure A' of Foissner [14]. General terminology follows Foissner and Berger [6], Gabilondo and Foissner [2], and Oertel et al. [7].

### 2.2. DNA Extraction, PCR Amplification and Sequencing

Five cells of *Fuscheriides baugilensis* sp. nov. were isolated from the raw culture using microcapillary under the stereomicroscope and were checked under the light microscope. The cells were transferred to the habitat water filtered by a 0.2 μm syringe filter (Minisart® CA Syringe Filters; Sartorius, Aubagne, France) at least five times to remove other eukaryotes. Each cell was then transferred to a 1.5 mL tube with a minimum amount of water using a microcapillary. Genomic DNA was extracted using a RED-Extract-*N*-Amp Tissue PCR Kit (Sigma, St. Louis, MO, USA). The PCR conditions were as follows: initial denaturation at 94 °C for 1 min 30 s, followed by 40 cycles of denaturation at 98 °C for 10 s, annealing at 58.5 °C for 30 s, and extension at 72 °C for 3 min, and a final extension step at 72 °C for 7 min. A slightly modified version of the primer New Euk A [15,16] and the primer LSU rev4 [17] were used to cover nearly the entire 18S rRNA gene. A MEGAquickspin Total Fragment DNA Purification Kit (iNtRON Biotechnology, Korea) was used to purify of the PCR products. DNA sequencing was performed using the New Euk A and LSU rev4 primers, three internal primers (18SF790v2: 5′-AAA TTA KAG TGT TYM ARG CAG-3′, 18SR300: 5′-CAT GGT AGT CCA ATA CAC TAC-3′, and 18SF1470: 5′-TCT GTG ATG CCC TTA GAT GTC-3′), and an ABI 3700 sequencer (Applied Biosystems, Foster City, CA, USA). Sequence fragments were assembled using Geneious Prime 2019.2.3 [18].

### 2.3. Phylogenetic Analysis

The SSU rRNA gene sequences of *Fuscheriides baugilensis* sp. nov. was used in the phylogenetic analyses with 77 ciliates retrieved from the NCBI database, including three metopids as outgroup taxa: *Clevelandella panesthiae* (KC139719), *Metopus palaeformis* (AY007450), and *Nyctotherus ovalis* (AJ222678). The sequences were aligned using ClustalW [19] and both ends were manually trimmed in BioEdit 7.0.9.0 [20]. The length of the final alignment was 1609 bp. The best-fit model of substitution for phylogenetic analysis, TVM + I (0.4710) + G (0.3460) based on the Akaike information criterion (AIC), was selected using jModelTest 2.1.10 [21,22]. IQ-Tree 1.6.12 [23] was used to render maximum likelihood (ML) trees, with 1000 bootstrap replicates. MrBayes 3.2.7 [24] was used for Bayesian inferences (BI) analyses with Markov chain Monte Carlo (MCMC) for 3,000,000 generations with a sampling frequency of every 100 generations and the first 25% of trees were discarded as burn-in. Phylogenetic trees were visualized using the software package FigTree v1.4.4 (http://tree.bio.ed.ac.uk/software/figtree/). Pairwise distances were calculated in Mega 6.06 [25], using the *p*-distance method.

## 3. Results

### 3.1. Systematics

Subclass Haptoria Corliss, 1974
Order Haptorida Corliss, 1974
Suborder Acropisthiina Foissner and Foissner, 1988
Family Fuscheriidae Foissner et al., 2002
Genus *Fuscheriides* Foissner and Gabilondo in Gabilondo and Foissner, 2009
*Fuscheriides baugilensis* sp. nov.
Figures 1A–H and 2A–I.
ZooBank registration number: urn:lsid:zoobank.org:act:6B0150FA-FF41-43C3-8598-F4609141323D.

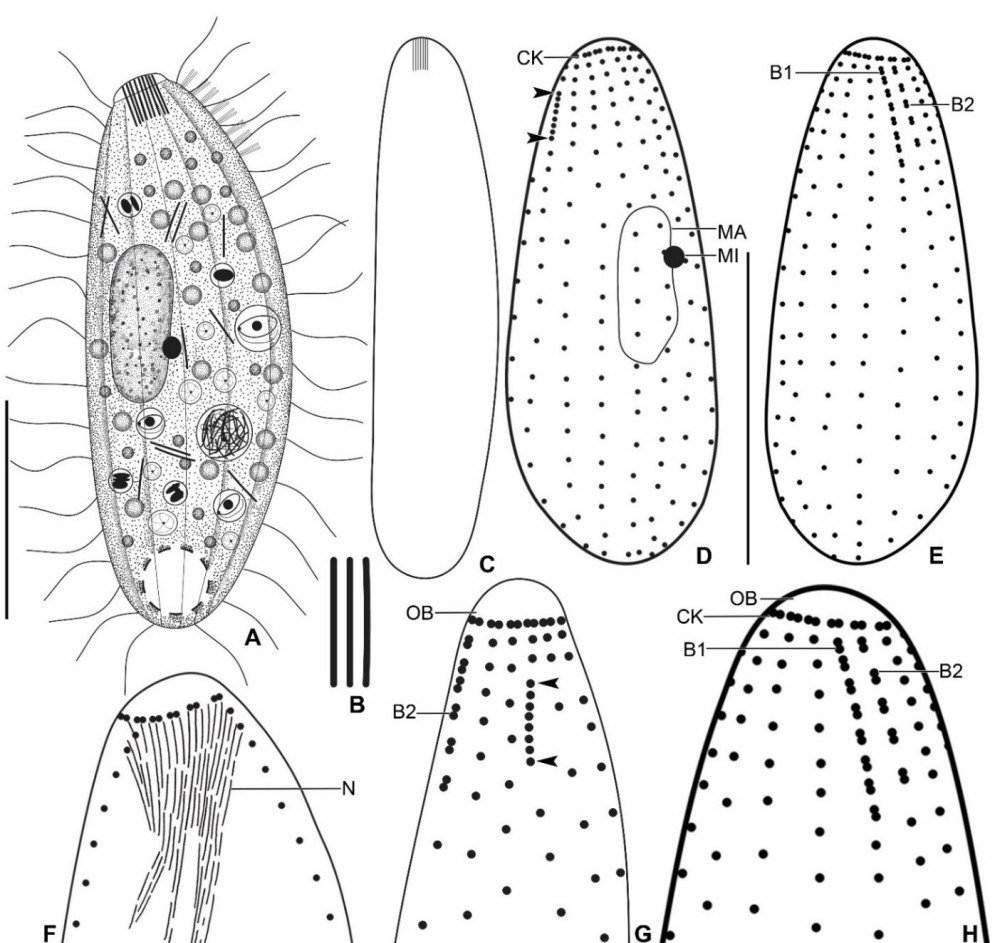

**Figure 1.** *Fuscheriides baugilensis* sp. nov. from life (**A–C**) and after protargol impregnation (**D–H**). (**A**). A representative specimen showing the body shape, the rod-shaped extrusomes and the ellipsoidal macronucleus. (**B**). Extrusomes in vivo. (**C**). Outline of slenderly ellipsoid, starved specimen. (**D,E,G,H**). Ventral (**D**), dorsal (**E,H**), and right lateral (**G**) view of the holotype (**D,E,H**) and a paratype (**G**) specimen, showing the somatic ciliary rows, the dikinetidal circumoral kinety, the two dorsal brush rows, and the subapical condensation (arrowheads). (**F**). The nematodesmata originate from the circumoral dikinetids and the anterior somatic kinetids. CK, circumoral kinety; B1–2, dorsal brush rows; MA, macronucleus; MI, micronuclei; N, nematodesmata; OB, oral bulge. Scale bars: 20 µm.

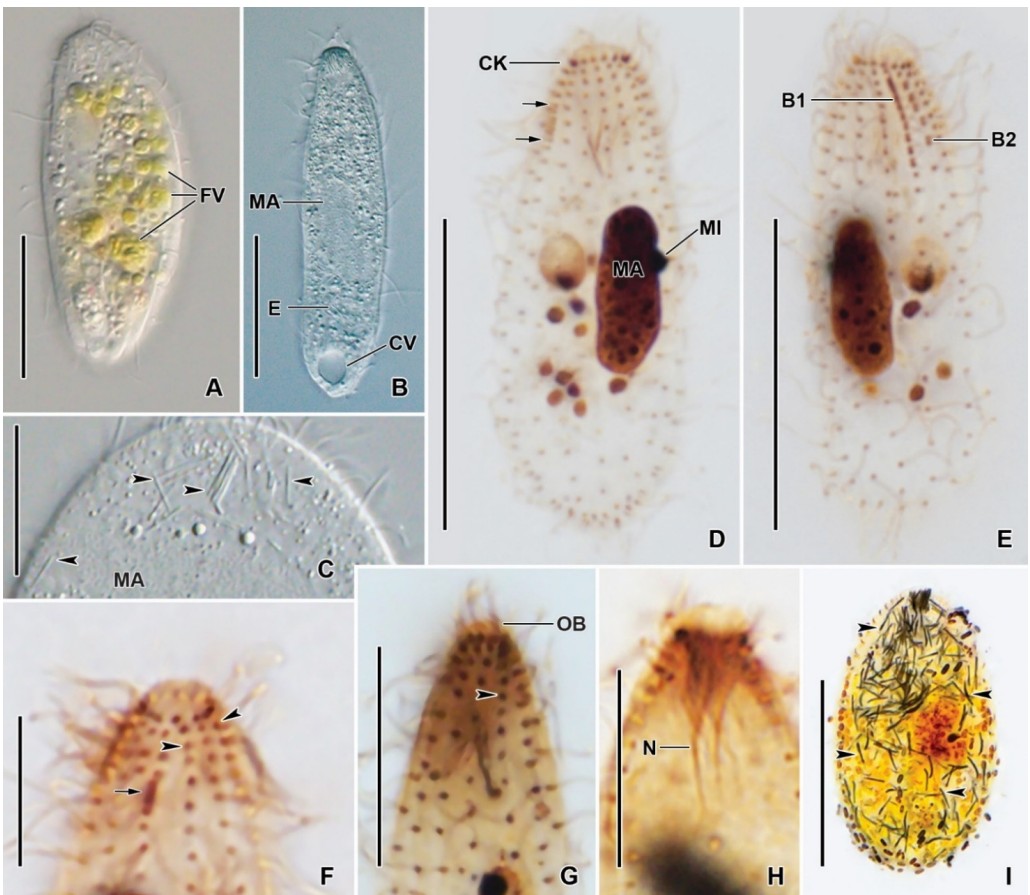

**Figure 2.** *Fuscheriides baugilensis* sp. nov. from life (**A**–**C**) and after protargol (**D**–**H**) and silver carbonate impregnation (**I**). (**A**,**B**). Lateral views of well-fed (**A**) and starved specimen (**B**), showing the body outline, the ellipsoid macronucleus, the food vacuoles containing green algae and flagellates, the oral and cytoplasm extrusomes, and the contractile vacuole. (**C**). Optical section showing the rod-shaped extrusomes (arrowheads). (**D**,**E**). Ventral and dorsal view of the holotype specimen, showing the somatic ciliature, the nuclear apparatus, and the subapical condensation (arrows) in the right body side. (**F**,**G**). Right side (**F**) and ventral (**G**) view, showing the subapical condensation (arrow) and the anteriorly curved to left side kineties (arrowheads). (**H**). Optical section showing the nematodesmata originating from the circumoral kinety and anterior somatic kinetids and form an obconical oral basket. (**I**). Developing extrusomes (arrowheads). B1–2, dorsal brush rows; CV, contractile vacuole; CK, circumoral kinety; E, extrusomes; FV, food vacuoles; MA, macronucleus; MI, micronuclei; N, nematodesmata; OB, oral bulge. Scale bars: 20 μm (**A**,**B**,**D**,**E**,**I**) and 10 μm (**C**,**F**–**H**).

### 3.2. Species Diagnosis

The body size was 30–55 × 15–20 μm in vivo and 21–36 × 11–17 μm after protargol impregnation. The body length:width ratio was approximately 1.8–2.8:1. The body shape was ovate to oblong and slightly curved. There was macronuclear nodule ellipsoidal in or anterior to the mid-body. The micronucleus globular was attached to the macronucleus. Extrusomes were oblong to rod-shaped, 3–5 × 0.3 μm. 14–16 ciliary rows. Two isostichad dorsal brush rows were present. Single subapical ciliary condensation right of the dorsal brush was composed of 5–8 densely arranged kinetids and was 1.8–3.9 μm long. Nematodesmata originated from circumoral dikinetids and oralized somatic monokinetids and extended in the anterior quarter of the body, forming an obconical oral basket.

### 3.3. Type Locality

Temporary puddle (after rainfall) on a footpath (Baugil) behind the Gangneung-Wonju National University, Gangneung, Korea (N 37°46′30.0″ E 128°51′46.8″).

### 3.4. Type Material

The slide containing the holotype (NNIBRPR21232) and one paratype slide (NNI-BRPR21233) with protargol-impregnated specimens were deposited at the Nakdonggang National Institute of Biological Resources (NNIBR), Sangju, Korea.

### 3.5. Etymology

The species was named after the footpath name in which it was discovered, i.e., the famous Baugil path in Gangwon-do province.

### 3.6. Description

The cell size was approximately 30–55 × 15–20 μm in vivo, and 21–36 × 11–17 μm after protargol impregnation (Table 1). The body was ovate to oblong, widened in the mid-body and anteriorly curved ventrally and slightly to left side; slightly narrowed and truncated anteriorly, and widely rounded posteriorly (Figures 1A,C and 2A,B). The cortex was flexible and furrowed along somatic ciliary rows, and non-contractile and cortical granules were lacking (Figures 1A and 2A,B). The macronucleus was ellipsoidal in vivo and globular to slenderly ellipsoidal after protargol impregnation, in or near the mid-body, 6–12 × 4–7 μm in size after protargol impregnation. The micronucleus globular, attached to the macronucleus, was 1.1–2.7 × 1.1–2.3 μm after protargol impregnation (Figures 1A,D and 2A–E, Table 1). Contractile vacuole was located in the posterior body end, was about 5 μm in diameter, and had about three excretory pores (Figures 1A and 2B). Extrusomes were oblong to rod-shaped, 3–5 × 0.3 μm in vivo, formed a bundle in the oral bulge and scattered in the cytoplasm. Developing extrusomes were 3–5 μm long and became thicker, sometimes acicular and curved after silver carbonate impregnation (Figures 1A,B and 2B,C,I). Cytoplasm hyaline, packed with lipid droplets 1–3 μm in diameter and many food vacuoles of up to 8 μm across were filled with green algae and flagellates (Figures 1A and 2A–C). The species swims fast by rotating about main body axis and never rests.

Cilia were about 8 μm long in vivo; 14–16 meridional monokinetidal somatic ciliary rows, and the spacing of kinetids gradually increased to the mid-body and then gradually decreased again posteriorly; mid-ventral kinety with 16–23 monokinetids (Figures 1A,D,E,G,H and 2D–G, Table 1). Two to four ventral kineties (right to subapical ciliary condensation) anteriorly slightly curved leftward (Figure 2F,G). Two rows differentiated anteriorly to the isostichad (dorsal brush row length difference <20%) dorsal brush; brush row 1 composed of 7–10 dikinetids, about 5.9 μm long; and brush row 2 composed of 5–6 dikinetids, about 5.0 μm long; space between dikinetids gradually increase posteriorly, brush bristles rod-shaped and 2–3 μm long in vivo (Figures 1E,G,H and 2E, Table 1). Subapical ciliary condensation was composed of 5–8 monokinetids, about 2.8 μm long, in the third kinety right of brush row 2, and were separated from circumoral dikinetids by two somatic kinetids (Figures 1D,G and 2D,F, Table 1).

The oral bulge was discoidal, indistinct in vivo and recognizable after protargol impregnation, about 1.2 μm high (Figures 2A,D–H and 2E–H). Circumoral kinety at the base of the oral bulge was composed of 14–16 transversely arranged dikinetids (Figures 1D–H and 2D–G). Nematodesmata were recognizable after protargol impregnation, originating from circumoral dikinetids and the anterior basal bodies of somatic kineties, and forming an indistinct, obconical oral basket about 9 μm long (Figures 1F and 2H, Table 1).

**Table 1.** Morphometric data on *Fuscheriides baugilensis* sp. nov.

| Characteristics [a] | Mean | M | SD | SE | CV | Min | Max | *n* |
|---|---|---|---|---|---|---|---|---|
| Body, length | 29.0 | 28.9 | 3.6 | 0.8 | 12.4 | 21.3 | 36.2 | 22 |
| Body, width | 13.1 | 13.1 | 1.5 | 0.3 | 11.6 | 10.5 | 16.8 | 22 |
| Body length:width, ratio | 2.2 | 2.2 | 0.3 | 0.1 | 13.1 | 1.8 | 2.8 | 22 |
| Oral bulge, height | 1.2 | 1.2 | 0.2 | 0.0 | 13.9 | 0.9 | 1.5 | 14 |
| Oral bulge, width | 3.5 | 3.4 | 0.5 | 0.1 | 13.1 | 2.5 | 4.5 | 23 |
| Body width:oral bulge width, ratio | 3.6 | 3.6 | 0.3 | 0.1 | 8.8 | 2.9 | 4.2 | 14 |
| Circumoral kinety to macronucleus, distance | 8.5 | 7.9 | 2.9 | 0.6 | 34.7 | 4.9 | 17.0 | 21 |
| Macronucleus, length | 8.3 | 8.0 | 1.5 | 0.3 | 18.5 | 6.0 | 11.6 | 22 |
| Macronucleus, width | 5.4 | 5.3 | 0.7 | 0.2 | 13.2 | 4.3 | 6.9 | 22 |
| Macronucleus length:width, ratio | 1.6 | 1.5 | 0.4 | 0.1 | 23.7 | 1.1 | 2.7 | 22 |
| Macronucleus, number | 1.0 | 1.0 | 0.0 | 0.0 | 0.0 | 1.0 | 1.0 | 23 |
| Micronucleus, length | 2.0 | 1.9 | 0.4 | 0.1 | 21.6 | 1.1 | 2.8 | 18 |
| Micromucleus, width | 1.6 | 1.5 | 0.4 | 0.1 | 22.0 | 1.1 | 2.3 | 18 |
| Micronucleus, number | 1.0 | 1.0 | 0.0 | 0.0 | 0.0 | 1.0 | 1.0 | 18 |
| Somatic kineties, number | 15.2 | 16.0 | 1.0 | 0.2 | 6.6 | 14.0 | 16.0 | 23 |
| Circumoral dikinetids, number | 15.2 | 16.0 | 1.0 | 0.2 | 6.6 | 14.0 | 16.0 | 23 |
| Kinetids in mid-ventral kinety, number | 18.2 | 18.0 | 2.2 | 0.5 | 11.9 | 16.0 | 23.0 | 17 |
| Dorsal brush row 1, length | 5.9 | 6.0 | 0.7 | 0.2 | 11.7 | 4.4 | 6.6 | 11 |
| Dorsal brush row 1 dikinetid, number | 8.3 | 8.0 | 0.9 | 0.3 | 10.9 | 7.0 | 10.0 | 11 |
| Dorsal brush row 2, length | 4.9 | 4.9 | 0.8 | 0.2 | 15.8 | 4.0 | 6.2 | 11 |
| Dorsal brush row 2 dikinetid, number | 5.5 | 5.0 | 0.5 | 0.2 | 9.6 | 5.0 | 6.0 | 11 |
| Dorsal brush rows, number | 2.0 | 2.0 | 0.0 | 0.0 | 0.0 | 2.0 | 2.0 | 11 |
| Subapical condensation, length | 2.8 | 2.6 | 0.6 | 0.1 | 19.9 | 1.8 | 3.9 | 21 |
| Kinetids in subapical condensation, number | 6.4 | 7.0 | 0.9 | 0.2 | 13.5 | 5.0 | 8.0 | 21 |
| Kinetids anterior to subapical condensation, number | 2.0 | 2.0 | 0.0 | 0.0 | 0.0 | 2.0 | 2.0 | 21 |
| Nematodesmata, length | 9.2 | 9.0 | 1.2 | 0.3 | 13.3 | 7.7 | 11.5 | 14 |

[a] Data based on protargol-impregnated specimens. All measurements in μm. CV—coefficient of variation in %, M—median, Max—maximum, Mean—arithmetic mean, Min—minimum, *n*—number of specimens investigated, SD—standard deviation, SE—standard error of arithmetic mean.

*3.7. Phylogenetic Analysis of Fuscheriides baugilensis sp. nov.*

The SSU rDNA sequence of *Fuscheriides baugilensis* sp. nov. is 1488 base pairs long, has a GC content of 39%, and is available under GenBank accession number OM291840. Phylogenetic trees using ML and BI analyses show rather similar topologies, thus only the ML tree is presented with both the bootstraps (ML) and the posterior probabilities (BI) are included (Figure 3). According to the new phylogenetic tree, the family Fuscheriidae is paraphyletic and consists of two fully supported subclades. The first subclade consists of four sequences: *F. baugilensis* sp. nov. and *Pseudofuscheria terricola*, which cluster together with high supporting values (98 ML, 0.99 BI) and exhibit a similarity of 99.2% (12 nucleotides difference). Both sequences cluster with an unidentified *Fuscheria* population (*Fuscheria* sp.; JF263448) with low support (67 ML, 0.68 BI). Together, these three sequences form a sister clade to an unidentified *Fuscheriides* population (*Fuscheriides* sp.; MG264144) with full support. The second subclade consists of *Fuscheria nodosa*, *Fuscheria uluruensis* and an unidentified *Enchelyodon* population.

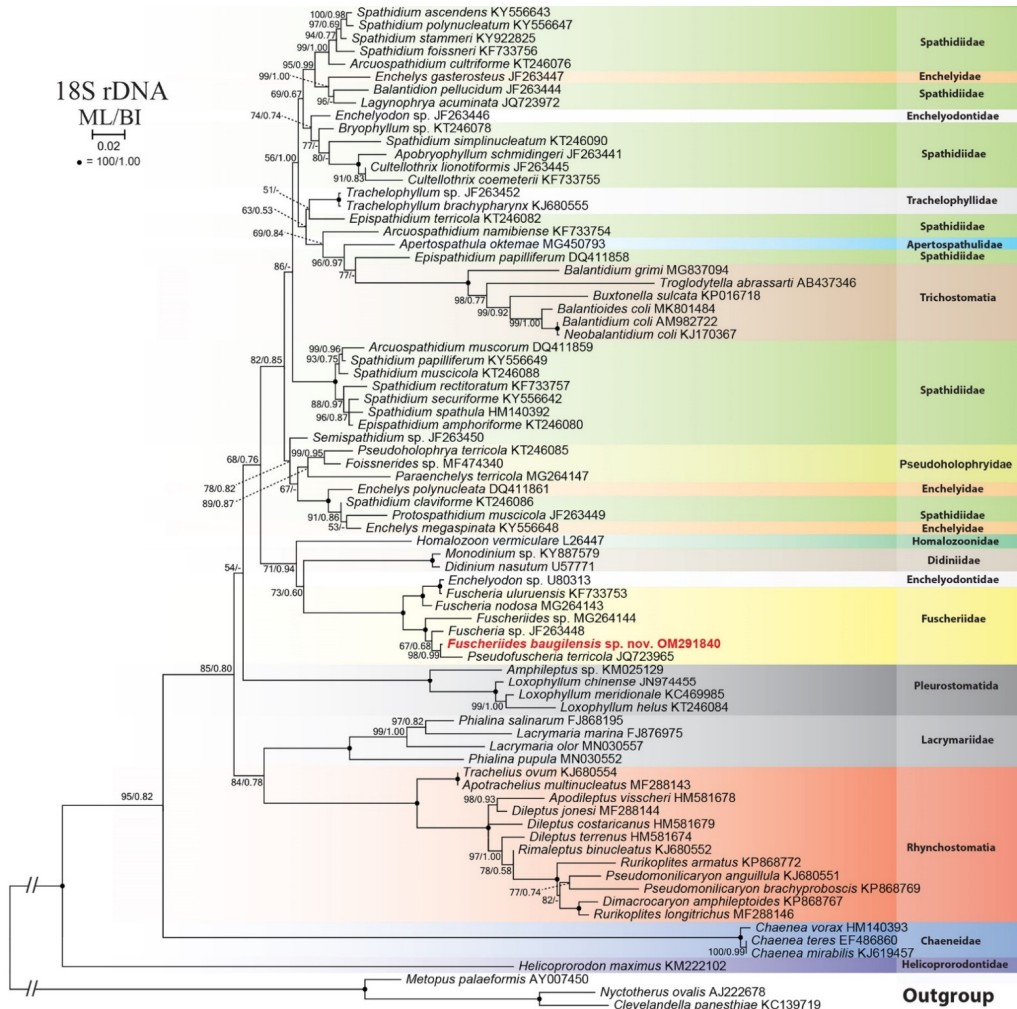

**Figure 3.** Maximum likelihood (ML) tree based on 18S rRNA gene sequences, showing the phylogenetic position of *Fuscheriides baugilensis* sp. nov. Newly obtained sequence is in bold. GenBank accession numbers follow species names. Numbers at the nodes represent the maximum likelihood (ML) bootstrap values and the Bayesian inference (BI) posterior probabilities. Dashes indicate bootstrap values < 50%, posterior probabilities < 0.5, or different topologies in BI and ML phylogenies. The scale bar represents two nucleotide substitutions per 100 nucleotides.

## 4. Discussion

### 4.1. Morphological Comparison of Fuscheriides baugilensis sp. nov. with Similar Species

Up to date, ten genera have been assigned to the family Fuscheriidae based on the number of dorsal brush rows, the shape and location of the extrusomes, and the presence/absence of the subapical ciliary condensation as follows: *Aciculoplites* Foissner and Gabilondo in Gabilondo and Foissner, 2009 [2] (with two brush rows and acicular extrusomes); *Actinorhabdos* Foissner, 1984 [26] (with two brush rows and graver-shaped extrusomes); *Apocoriplites* Oertel et al., 2008 [7] (with two brush rows and without extrusomes); *Coriplites* Foissner, 1988 [5] (with three brush rows and without extrusomes); *Diplites* Foissner, 1998 [27] (with two brush rows and ellipsoidal or clavate oral extrusomes and rod-shaped somatic extrusomes); *Dioplitophrya* Foissner et al., 2002 [1] (with three brush rows and pin-shaped and clavate extrusomes in oral bulge and cytoplasm); *Fuscheria* Foissner, 1983 [28], type genus (with two brush rows and pin-shaped extrusomes); *Fuscheriides* Foissner and Gabilondo in Gabilondo and Foissner, 2009 [2] (with two brush rows, oblong extrusomes, and a subapical ciliary condensation); *Pseudofuscheria* Foissner and Berger, 2021 [6] (with two brush rows, pin-shaped extrusomes, and subapical ciliary

condensations); and *Renoplites* Foissner, 2016 [29] (with two brush rows and reniform extrusomes) [1,2,5–7,26–31].

*Fuscheriides* and *Pseudofuscheria* are the only fuscheriid genera with subapical ciliary condensation. The two genera are similar in having only two dorsal brush rows and can only be distinguished based on the extrusomes shape. *Fuscheriides baugilensis* sp. nov. differs from the type species and sole congener, *F. tibetensis* Foissner and Gabilondo in Gabilondo and Foissner, 2009, by the number of somatic ciliary rows (14–16 vs. invariably 7), the length of the extrusomes (3–5 vs. 2 µm), and the shape of the macronucleus (ellipsoidal vs. reniform) (Table 2). *Pseudofuscheria terricola* (Berger et al. 1983) Foissner and Berger, 2021 [6] is the most similar species to *F. baugilensis* sp. nov. Both species have a similar shape and number of somatic ciliary rows (12–19 vs. 14–16). However, the two species can be easily separated by the shape of the extrusomes (nail-shaped vs. oblong to rod-shaped). Moreover, *Pseudofuscheria terricola* is larger than *F. baugilensis* sp. nov. (80–100 µm vs. 30–55 µm in vivo), has one or two (vs. invariably one) subapical condensations and possesses a higher number of kinetids in ventral kinety (12–45 vs. 16–23) [1,2,6,30].

**Table 2.** Comparison of *Fuscheriides baugilensis* sp. nov. with closely related species.

| Characteristics | *Fuscheriides baugilensis* sp. nov. | *Fuscheriides tibetensis* | *Pseudofuscheria terricola* | *P. magna* | *Fuscheria nodosa nodosa* | *Fuscheria nodosa salisburgensis* | *Fuscheria uluruensis* |
|---|---|---|---|---|---|---|---|
| Body, length (µm) | 21–43 | 33–62 | 44–78 | 85–145 | 35–46 | 82–137 | 65–87 |
| Body, width (µm) | 10–18 | 5–17 | 12–28 | 33–95 | 18–26 | 45–97 | 45–62 |
| Macronucleus, numbers | 1 | 1 | 1 | 1 | 1 | 1 | 8–28 |
| Macronucleus, shape | Ellipsoidal | Reniform | Horseshoe-shaped | Oblong, curved oblong, or horseshoe | Horseshoe-shaped | Strand-shaped | Elongate ellipsoidal |
| Somatic kineties, number | 14–16 | 7 | 12–19 | 25–34 | 24–28 | 42–45 | 42–50 |
| Kinetids in ventral kinety, number | 16–23 | 11–27 | 12–45 | 40–80 | 20–35 | 31–66 | 41–88 |
| Brush rows, numbers | 2 | 2 | 2 | 2 | 2 | 2 (rarely 3) | 2 |
| Subapical condensation | Present | Present | Present | Present | Absent | Absent | Absent |
| Subapical condensation rows, number | 1 | 1 | 1 or 2 | 2 | - | - | - |
| Kinetids in front of condensation, number | 2 | 3 | 4 | 5–6 | - | - | - |
| Extrusomes, shape | Oblong to rod-shaped | Oblong | Nail-shaped | Nail-shaped | Nail-shaped | Nail-shaped | Nail-shaped |
| Extrusomes, size (µm) | ~4 × 0.3 | ~2 × 0.3 | ~4–7 | | ~10 | ~10 | ~9–15 |
| Habitat, country | Temporary pond after rainfall, Korea | Salty vegetation soil (10‰), South Tibet | Soil, Austria | Floodplain soil, Australia | Pond, Australia | Soil, Austria | Soil, Austria |
| Reference | Present study | [2] | [30] | [6] | [31] | [2] | [2] |

*4.2. Phylogenetic Analyses*

The new phylogenetic tree agrees with previous studies in that the family Fuscheriidae is paraphyletic [32–35]. However, the family is still underrepresented in the phylogenetic tree and there are only a few species belonging to three out of ten genera with the available molecular data. The available fuscheriid sequences form a clade made of two subclades with full support. *Fuscheriides baugilensis* sp. nov. nests in a subclade containing *Pseudofuscheria terricola* and two other unidentified species likely belong to the genera *Fuscheriides* and *Pseudofuscheria*. The close relationship of *Fuscheriides* and *Pseudofuscheria* is supported by the presence of the subapical ciliary condensation, which is used as the sole generic feature that differentiates *Fuscheria* from *Pseudofuscheria* [6]. The other subclade contains *Fuscheria nodosa*, *F. uluruensis*, and unidentified *Enchelyodon* species without morphological data and thus misidentification cannot be excluded. Interestingly, both *Fuscheria* and *Pseudofuscheria* have nail-shaped extrusomes but they nest in different subclades, suggesting that the shape of the extrusomes has a lower taxonomic value than suggested by previous studies [1,2,6,7,29,30], i.e., only a species-specific character as in other litostomatean families, for instance, Spathidiidae and Trachelophyllidae [1,29,36,37]. Based on this assumption, the fuscheriid species should be assigned into only three groups (genera): (1) species with two brush rows; (2) species with three brush rows; and (3) species with two brush rows and ciliary condensations. However, more molecular data are needed on the other genera to test the value of each generic character.

**Author Contributions:** Conceptualization, S.W.J., A.O., S.W.N. and J.-H.J.; Data curation, A.O.; Funding acquisition, J.-H.J.; Investigation, S.W.J. and A.O.; Methodology, S.W.J., J.-H.J. and A.O.; Visualization, S.W.J. and A.O.; writing—original draft, S.W.J., A.O. and S.W.N.; writing—review & editing, A.O., J.-H.J. and S.W.N. All authors have read and agreed to the published version of the manuscript.

**Funding:** This study was supported by a grant from the Nakdonggang National Institute of Biological Resources (NNIBR) of Korea (NNIBR202201105).

**Data Availability Statement:** The data presented in this study can be found in online repositories. The names of the repository/repositories and accession number(s) can be found in the article.

**Acknowledgments:** We thank the anonymous reviewers and the editor who made constructive and invaluable suggestions and comments.

**Conflicts of Interest:** The authors declare no conflict of interest.

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
