# Peer review of "Morphology and Molecular Phylogeny of Fuscheriides baugilensis sp. nov. (Protozoa, Ciliophora, Haptorida) from South Korea†"

_diversity, doi:10.3390/d14020070_

Round 1

Reviewer 1 Report

This study describes a new species of a poorly known genus Fuscheriides. The methods used are appropriate, the results are clearly presented, the conclusions are supported by the results, the relevant literature has been cited and the illustrations are of good quality. There are a few points that need to be addressed. These are highlighted in the notes annotated in the manuscript. Here is one major comment:

1. Fusheriides (Gabilondon and Foissner, 2009) has some ventro-lateral diliary rows curved leftwards anteriorly, but the present species does not. Please check the slides carefully. I think it is likely a member of Pseudofusheria. Your phylogeny analyses also support it.

Author Response

1. Fusheriides (Gabilondon and Foissner, 2009) has some ventro-lateral diliary rows curved leftwards anteriorly, but the present species does not. Please check the slides carefully. I think it is likely a member of Pseudofusheria. Your phylogeny analyses also support it.
Answer: We fixed this part in the manuscript. Our species is small and it is difficult to recognize the anteriorly curved somatic kineties but after more and careful observations on our preparations, we could recognize this character and provided micrographs in figure 2. 
Also you mentioned that this species is likely a member of the genus Pseudofuscheria. However, this is probably right but based on our current knowledge, we cannot assign this species to the genus Pseudofuscheria because the shape of the extrusomes is one of the most important characters used by Foissner in his publications on the fuscheriid genera and he characterized the genera Fuscheria and Pseudofuscheria by the pin-shaped extrusomes (vs. oblong to rod-shaped in Fusheriides baugilensis). Also, the fuscheriids are underrepresented in the phylogenetic tree and most genera and the type species of Fuscheriides and Pseudofuscheria lack DNA sequences. Possibly, future investigations on the fuscheriids will show that the shape of extrusomes is only a species specific character.

2. Lines 18-19 in the abstract: I cannot understand this sentence. what does 'taxonomic value' mean? please reword it.
Answer: We rewrote the sentence explaining that the position of the sequences in the phylogenetic tree suggest that the subapical ciliary condensation has a higher taxonomic value than the extrusome shape in genera separation.

3. Reviewer's comments on the pdf file 

Answer: We have revised our manuscript according to the comments.

Reviewer 2 Report

The presented paper focuses on the description of new species belonging to the Fuscheriidae family. Authors not only describe morphological characterisation but also present molecular phylogeny based on the SSU rDNA sequences, obtained from single cells of the new species. 

The article is well written, methods and results description is accurate. Obtained results are well discussed with literature data.  

However, please explain why burn-in in BI analysis is 7500 where the standard for phylogenetic analysis is 25%. Is it a typo? If this is the case where burn-in was determined specifically for the output data then used for this purpose programs should be mentioned in that section. 

The holotype and paratype deposition numbers as well as Genbank accession numbers for SSU rDNA sequences for new species should be filled.

Author Response

1. Comment: However, please explain why burn-in in BI analysis is 7500 where the standard for phylogenetic analysis is 25%. Is it a typo? If this is the case where burn-in was determined specifically for the output data then used for this purpose programs should be mentioned in that section. 
Answer: We have followed the standard. To avoid the confusion, however, we slightly modified the sentence. We calculated the 25% burn-in as follows: ngen = 3,000,000 generations and samplefreq = 100, the total samples are (3,000,000/100) = 30,000. Then the 25% burn-in should be 7,500.

2. Comment: The holotype and paratype deposition numbers as well as Genbank accession numbers for SSU rDNA sequences for new species should be filled.
Answer: the holotype, paratype, and GenBank accession numbers are provided in the revised version of the manuscript. Also, the Zoobank registration is also provided.

Reviewer 3 Report

Review of the manuscript Diversity-1565148 entitled: “Morphology and Molecular Phylogeny of Fuscheriides baugili sp. nov. (Protozoa, Ciliophora, Haptorida) from South Korea”

The submitted manuscript is an original research article that presents an investigation of the morphology and molecular phylogeny of a new haptorid ciliate, Fuscheriides baugili sp. nov., discovered in a temporary pond from South Korea. In my opinion, the introduction is well written, and clearly introduces the reader to the subject matter presented here. The methodology has been selected and presented correctly (slight comments below).

Materials and Methods

Did the Authors measure the DNA concentration after isolation? If so, please provide details.

Information on PCR conditions is missing. Please complete it.

It is not clear whether you’ve slightly modified the primers or did you used slightly modified primers (citation missing). Please clarify.

There is no detailed information about sequencing primers (citation). Are they contain PCR primers?

Results

Zoobank record number and GenBank record number are missing. Please provide them.

Discussion

I am aware that this is not the topic of this paper, but I strongly recommend future studies based on more morphological and molecular data as well as more haptorid ciliates samples. This will give a chance to verify the status of the Fuscheriidae family.

Despite the above reservations, I believe that it is a very valuable text which in revised form can be published in Diversity.

Author Response

Materials and Methods
1. Comment: Did the Authors measure the DNA concentration after isolation? If so, please provide details.
Answer: We did not measure the DNA concentration after isolation, instead we extract DNA from five cells separately.

2. Comment: Information on PCR conditions is missing. Please complete it.
Answer: The PCR conditions are complete in lines 67-69.

3. Comment: It is not clear whether you’ve slightly modified the primers or did you used slightly modified primers (citation missing). Please clarify.
Answer: We used a slightly modified version of the primer Euk A in Medlin et al. (1988) (New Euk A in Jung et al. (2011)) and the primer LSU rev4 (Sonnenberg et al., 2007). The sentence is corrected in the manuscript.

4. Comment: There is no detailed information about sequencing primers (citation). Are they contain PCR primers?
Answer: We included the PCR primers as sequencing primers besides the three internal primers.

Results
5. Comment: Zoobank record number and GenBank record number are missing. Please provide them.
Answer: Zoobank record and GenBank accession number are provided.

Discussion
6. Comment: I am aware that this is not the topic of this paper, but I strongly recommend future studies based on more morphological and molecular data as well as more haptorid ciliates samples. This will give a chance to verify the status of the Fuscheriidae family.
Answer: We are collecting data on other litostomatean, especially haptorid, species to increase the sampling and solve the phylogenetic issues as the non-monophyly.